# PeerJ

# Using network clustering to predict copy number variations associated with health disparities

Yi Jiang[1,3], Hong Qin[2,3] and Li Yang[1]

[1] Department of Computer Science and Engineering, University of Tennessee at Chattanooga, TN, USA
[2] Departement of Biology, Spelman College, Atlanta, GA, United States
[3] Co-first authors.

## ABSTRACT

Substantial health disparities exist between African Americans and Caucasians in the United States. Copy number variations (CNVs) are one form of human genetic variations that have been linked with complex diseases and often occur at different frequencies among African Americans and Caucasian populations. Here, we aimed to investigate whether CNVs with differential frequencies can contribute to health disparities from the perspective of gene networks. We inferred network clusters from human gene/protein networks based on two different data sources. We then evaluated each network cluster for the occurrences of known pathogenic genes and genes located in CNVs with different population frequencies, and used false discovery rates to rank network clusters. This approach let us identify five clusters enriched with known pathogenic genes and with genes located in CNVs with different frequencies between African Americans and Caucasians. These clustering patterns predict two candidate causal genes located in four population-specific CNVs that play potential roles in health disparities

## INTRODUCTION

Health disparities refer to differences in the disease distribution and/or health outcomes across racial and ethnic groups. In the United States, health disparities in African Americans are found in life expectancy, death rates, and health measures (*National Center for Health Statistics, 2013*). In addition to social determinants such as socio-economical status, health care access and cultural practices, human genetic variations play a significant role in health disparities. Genetic variations at different frequencies among populations can lead to differences in disease susceptibility. Studies on genetic variations and disease association are greatly advanced by the completion of the International HapMap Project and new genome sequencing techniques (*Ramos & Rotimi, 2009*).

Genome-wide association studies (GWAS) are currently an effective approach to identify disease-associated genetic variations (*Hirschhorn & Daly, 2005*; *Wang et al., 2005*). Although GWAS have revealed many disease-associated single nucleotide polymorphisms

Corresponding author
Hong Qin, hqin@spelman.edu

(SNPs), GWAS are often limited to individual genetic variations and often do not address complex gene interactions. Moreover, associated SNPs are often located in haplotype blocks that contain more than one gene. To address these limitations, human gene networks have been used to improve GWAS detection of genes associated with complex diseases, such as the comorbidity analysis (*Sharma et al., 2013*), an improved guilt-by-association method (*Baranzini et al., 2009*; *Lee et al., 2011*), and a distance-based scoring method using seeded diseases genes (*Liu et al., 2012*).

Copy number variations (CNVs) are duplications or deletions of genomic segments that can contain one or more genes (*McCarroll & Altshuler, 2007*). CNVs have been associated with complex diseases such as autism (*Gilman et al., 2011*; *Glessner et al., 2009*). Computational tools and methods, such as the CNV annotator (*Zhao & Zhao, 2013*) and NETBAG (*Gilman et al., 2011*), have been developed to address the potential roles of CNVs in human diseases. Recently, it was reported that CNVs can occur at different frequencies between African Americans and Caucasians (*McElroy et al., 2009*), and naturally the question about the potential roles of CNVs in health disparity is raised.

Here, we aim to investigate the clustering of pathogenic genes and genes in CNVs with different population frequencies in two human gene/protein networks, in order to better understand health disparities between African Americans and Caucasians. The current human gene/protein networks contain thousands of interacting molecules (*Barabasi, Gulbahce & Loscalzo, 2011*; *Vidal, Cusick & Barabasi, 2011*). We will partition gene networks into clusters and use these clusters to predict potential diseases associated with population-specific CNVs, based on the rationale that interacting genes often share similar functions (*Pizzuti, Rombo & Marchiori, 2012*).

## MATERIALS AND METHODS

Our overall work flow is shown in Fig. 1. To identify potential diseases associated with CNVs, our basic idea is to identify gene interaction clusters that involve genes in population-specific CNVs. The diseases associated with a CNV-gene's interacting genes are potential diseases associated with this CNV. Specifically, we first obtained two human gene/protein networks and partitioned them into gene clusters. We then performed statistical tests on each cluster to estimate its significances in containing pathogenic genes and genes in population-specific CNVs. Finally, we ranked gene clusters based on false discovery rates (FDRs). High-ranked clusters were enriched both for pathogenic genes and for genes in CNVs with differential frequencies between African-Americans and Caucasians. These clusters were then searched for enriched Gene Ontology (GO) terms and related disease phenotypes.

### Network clustering

We obtained two human gene/protein networks, one from Human Protein Reference Database (HPRD) (*Mishra et al., 2006*; *Peri et al., 2003*; *Prasad et al., 2009*) and another from MultiNet (*Khurana et al., 2013*). The HPRD network (referred to as HPRDNet) contains only physical protein–protein interactions (PPIs), whereas MultiNet is a unified network including PPI, phosphorylation, metabolic, signaling, genetic and regulatory

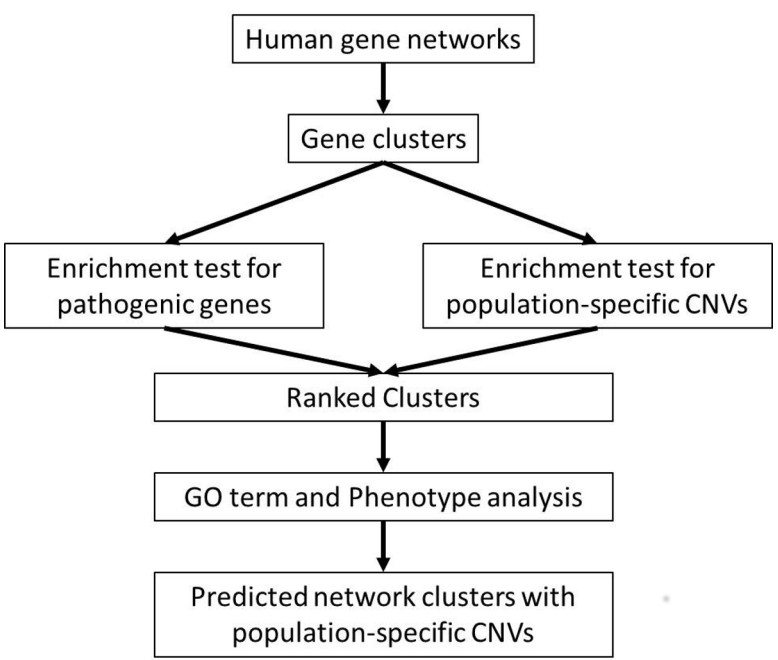

**Figure 1 Overview of our approach to identify CNVs associated with health disparities.**

networks. These two networks share 8,468 genes (89.6% of HPRDNet and 58.6% of MultiNet) but only 8,769 interactions (23.8% of HPRDNet and 8% of MultiNet). These two networks were both partitioned into gene clusters using the Markov Cluster (MCL) Algorithm (*Van Dongen, 2000*). Clustering was done with the inflation parameter I ranging from 1.1 to 2.0 with a step of 0.1. Descriptive statistics of the two networks and their clustering results are summarized in Table S1.

## Mapping of CNVs and SNPs

CNV coordinates were obtained from a CNV map in African Americans and Caucasians (*McElroy et al., 2009*). There are three types of CNVs in this map: (1) CNVs only occurred in African Americans; (2) CNVs only occurred in Caucasians; and (3) CNVs occurred in both African Americans and Caucasians. To simplify the analysis, we further partitioned the last type: CNVs that occurred more than 50% in African Americans or in Caucasians were combined with the first and second types of CNVs, respectively. This repartition resulted in two modified CNV sets with differential population frequencies. The coordinates of these CNVs were then searched in the UCSC Genome Database (*Karolchik et al., 2014*) through its MySQL API to obtain the corresponding gene sets. For simplicity, CNVs that occur more frequently in African Americans were called African-American CNVs or CNV_AA; CNVs that occur more frequently in Caucasians were called Caucasian CNVs or CNV_ CA.

Disease-associated SNPs were retrieved from a file, OmimVarLocusIdSNP.bcp, from the FTP site of Single Nucleotide Polymorphism Database (dbSNP) (*Sherry et al., 2001*). Coordinates of these SNPs were then queried against the MySQL API of the UCSC Genome

**Table 1** (A) Contingency table for Fisher's exact test on pathogenic genes. (B) Contingency table for Fisher's exact test on CNV genes.

**(A)**

|  | Pathogenic genes | Non-pathogenic genes | Total |
|---|---|---|---|
| Genes in this cluster | $q$ | $m - q$ | $m$ |
| Genes in other clusters | $Q - q$ | $N - Q - m + q$ | $N - m$ |
| Total | $Q$ | $N - Q$ | $N$ |

**(B)**

|  | CNV genes | Non-CNV genes | Total |
|---|---|---|---|
| Genes in this cluster | $s$ | $m - s$ | $m$ |
| Genes in other clusters | $S - s$ | $N - S - m + s$ | $N - m$ |
| Total | $S$ | $N - S$ | $N$ |

**Notes.**

For each cluster, contingency tables were constructed for right-tailed Fisher's exact Tests. (A) is for pathogenic significance test, and (B) is for tests of enrichment significance of CNV genes (CNV_AA or CNV_CA genes). $Q$ and $q$ are the number of pathogenic genes in the whole networks and that in current cluster, respectively. $N$ and $m$ are the number of genes in whole networks and that in current cluster, respectively. $S$ and $s$ are the number of CNV_AA or CNV_CA genes in the whole networks and that in current cluster, respectively.

Database to identify genes in which those SNPs are located. This identified gene set was termed as pathogenic genes. Details of gene mapping results are shown in Table S2.

## Cluster analyses

Clusters were obtained from both HPRDNet and MultiNet using MCL with a range of ten inflation parameters. For each cluster, contingency tables were constructed using the numbers of pathogenic genes and CNVs related genes (Tables 1(A) and 1(B)). Right-tailed Fisher's exact tests were applied to these contingency tables to calculate enrichment significance of pathogenic genes, and CNV_ AA or CNV_ CA genes, respectively. Based on obtained $p$-values, false discovery rates (FDRs) were calculated using the Robust FDR Routine (*Pounds & Cheng, 2006*). Fisher's exact tests and Robust FDR Routine were both performed in the R statistical environment (*R Development Core Team, 2013*). Ranking was applied to clusters with $p$-value $<0.10$ and FDR $<0.20$ in both enrichment tests for pathogenic genes and population-preferred CNVs genes. Assuming both enrichment tests are independent, the FDR values were multiplied to jointly rank the network clusters. The same cluster analysis procedure was applied to clustering results with different MCL inflation parameters.

For clarity, we focused our functional analyses on clusters that were consistently ranked at the first place with different MCL inflation parameter values.

## Biological significance analyses

Biological relevance of selected network clusters were analyzed by GOrilla (*Eden et al., 2009*) to search for enriched gene ontology (GO) terms. In GOrilla search, genes in the selected clusters were target genes, and all genes in the network were treated as background genes. To investigate the possible links of population-specific CNVs to heath disparities, we first identified significantly enriched GO terms that are associated with CNV_ AA or
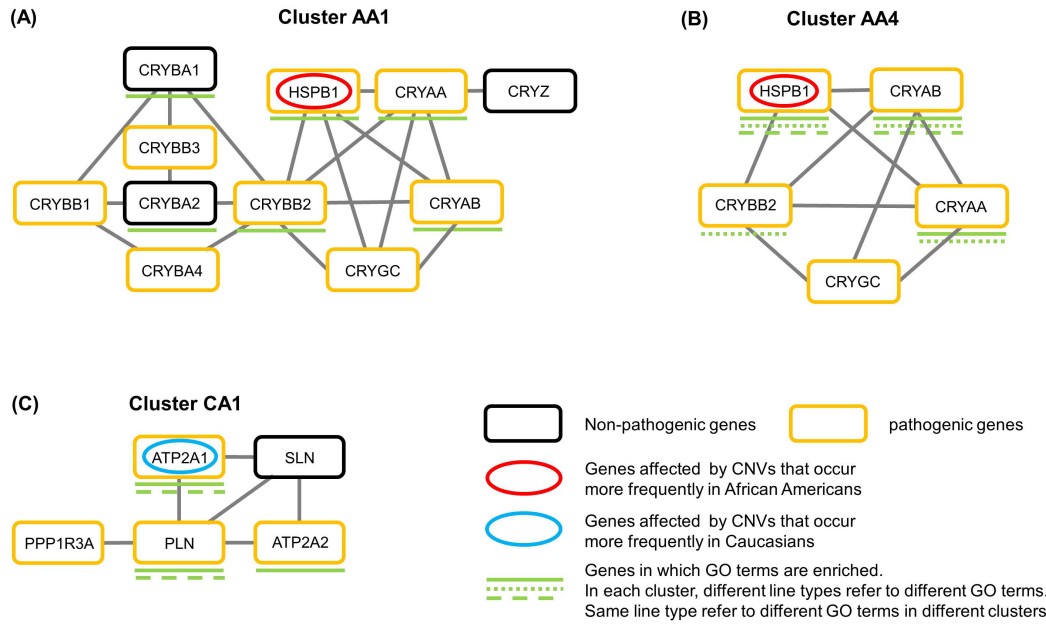

Figure 2 **Graph representations of selected clusters for biological significance analysis.** Each rounded rectangle represents a gene and each gray line represents a gene–gene interaction. Black rounded rectangles represent non-pathogenic genes and orange rounded rectangles represent pathogenic genes. Genes labeled with red or blue ovals are located in African American CNVs or in Caucasian CNVs. Genes with green lines share the same GO terms. In each cluster, different line types represent the enrichment of different GO terms. Line types shown in different clusters refer to the enrichment of different GO terms.

CNV˗ CA genes. We then focused on the pathogenic genes related to the enriched GO terms, and examined their associated disease phenotypes in OMIM database (*Online Mendelian Inheritance in Man, 2014*).

## RESULTS AND DISCUSSIONS

### Top-ranked network clusters

We performed cluster analyses with ten MCL inflation parameter values for both HPRDNet and MultiNet (Table S1), and scored the resulted clusters for their potential roles in CNV related health disparities (Table S3). For clarity, we focused on clusters that are consistently top-ranked with different MCL inflation parameters. The graph representations of selected clusters are shown in Fig. 2.

We found four similar clusters, (AA1, AA2, and AA3 in HPRDNet and AA4 in Multinet), that are enriched both for pathogenic genes and for genes located in African-American CNVs (Table 2). In HPRDNet, cluster AA1, AA2 and AA3 together were ranked at first place five times; and cluster AA4 were ranked five times in Multinet (Table S3). Cluster AA1 contains 11 genes, within which eight are pathogenic genes (Fig. 2A). Cluster AA2 and AA3 contain one and two more genes than cluster AA1, respectively (Fig. S1). In MultiNet, cluster AA4 contains five genes and can be considered as a sub-cluster of cluster AA1, AA2 and AA3 (Fig. 2B). In these four clusters, gene *HSPB1* is mainly duplicated in African Americans (Tables 2 and 3). Based on GO enrichment tests, this family of clusters

**Table 2  Cluster analysis results for HPRDNet and MultiNet.**

| Network | Cluster name | CNV_AA | CNV_CA | Pathogenic gene number | Cluster size |
|---|---|---|---|---|---|
| **HPRDNet** | AA1 | *HSPB1* | – | 8 | 11 |
| | AA2 | *HSPB1* | – | 8 | 12 |
| | AA3 | *HSPB1* | – | 8 | 13 |
| | CA1 | – | *ATP2A1* | 4 | 5 |
| **MultiNet** | AA4 | *HSPB1* | – | 5 | 5 |
| | CA1 | – | *ATP2A1* | 4 | 5 |

**Notes.**
Selected clusters were listed. CNV_AA and CNV_CA are CNV-related genes.

**Table 3  Detected genes with potential roles in health disparity and their located CNVs.**

| Gene | Chr | Gene coordinates | CNV region | CNV type | CNV occurrence preference |
|---|---|---|---|---|---|
| ***HSPB1*** | 7 | 75,931,861–75,933,614 | 75,867,431–76,481,102 | Duplication | Only in African American |
| | | | 75,929,740–76,481,102 | Duplication | Only in African American |
| | | | 75,929,740–76,568,388 | Duplication | More in African American than in Caucasian |
| *ATP2A1* | 16 | 28,889,726–28,915,830 | 28,306,730–28,936,772 | Duplication | Only in Caucasian |

**Notes.**
Chr represents chromosomes. CNV Regions are regions of CNVs identified in more than a single individual; all CNVs listed have a type of Duplication, referring to one copy increase. CNV Regions and Types are from the CNV map (*McElroy et al., 2009*). CNV Occurrence preference describes in which population those CNVs have higher occurrence frequency.

was found to be involved in visual perception and eye development. Since cluster AA1, AA2 and AA3 were selected from the same network and are highly similar to each other, only cluster AA1 and AA4 will be discussed further for their potential roles in health disparities.

In both HPRDNet and MultiNet, the same cluster, named as CA1, was identified to be enriched with both pathogenic genes and genes located in Caucasian CNVs (Table 2). Cluster CA1 was ranked at first place four times in HPRDNet and seven times in MultiNet (Table S3). This cluster contains five genes, and four of them are associated with diseases (Fig. 2C). The GO term enrichment tests suggested that cluster CA1 was involved in calcium ion transportation and muscle contraction regulation. Cluster CA1 contains gene *ATP2A1* that is duplicated only in Caucasians (Table 3).

## Duplication of *HSPB1* and health disparities in African Americans
Gene *HSPB1* is located in genomic duplication regions occurring more frequently in African Americans (Table 3), and is found in the cluster family of AA1, AA2, AA3, and AA4 (Table 2). For cluster AA1, only one GO molecular function term related to gene *HSPB1* is significantly enriched (Cluster AA1 in Table 4). For cluster AA4, in addition to the same enriched GO molecular functions term, three GO biological process terms and one GO cellu-

**Table 4  Enriched GO terms with CNV-genes in the identified network clusters.**

| Clusters | Involved genes | GO domain | GO ID | GO term |
|---|---|---|---|---|
| AA1 | HSPB1, CRYAA, CRYAB, CRYBB2, CRYBA1, CRYBA2 | Molecular function | GO:0042802 | Identical protein binding |
| AA4 | HSPB1, CRYAA, CRYAB | Biological process | GO:0043086 | Negative regulation of catalytic activity |
| | | Biological process | GO:0043066 | Negative regulation of apoptotic process |
| | | Biological process | GO:0043069 | Negative regulation of programmed cell death |
| | HSPB1, CRYAA, CRYAB, CRYBB2 | Molecular function | GO:0042802 | Identical protein binding |
| | HSPB1, CRYAB | Cellular component | GO:0030018 | Z disc |
| CA1[a] | ATP2A1, ATP2A2, PLN, SLN | Biological process | GO:0090257 | Regulation of muscle system process |
| | | Biological process | GO:0006816 | Calcium ion transport |
| | | Cellular component | GO:0033017 | Sarcoplasmic reticulum membrane |
| | ATP2A1, ATP2A2, PLN | Biological process | GO:0003012 | Muscle system process |
| | | Biological process | GO:0006874 | Cellular calcium ion homeostasis |
| | | Cellular component | GO:1902495 | Transmembrane transporter complex |
| | ATP2A1, ATP2A2, SLN | Cellular component | GO:0016529 | Sarcoplasmic reticulum |
| | ATP2A1, ATP2A2 | Biological process | GO:0032470 | Positive regulation of endoplasmic reticulum calcium ion concentration |
| | | Cellular component | GO:0031095 | Platelet dense tubular network membrane |

**Notes.**

Biological relevance of network clusters was analyzed by GOrilla (*Eden et al., 2009*) to search for enriched gene ontology (GO) terms. Genes in the selected clusters were used as target genes, and all genes in the networks were treated as background genes. Three types of GO terms were analyzed: biological process, molecular function and cellular component. The default $p$-value threshold ($1 \times 10^{-3}$) was used. In the results, enriched GO terms that are associated with CNV_ AA gene *HSPB1* and CNV_ CA gene *ATP2A1* were selected and listed in the table.

[a] When multiple enriched GO terms show similar meanings, we only presented the most general terms.

lar component term are found significantly enriched (Cluster AA4 in Table 4). In the genes with the enriched GO terms, four of them are known to be associated with diseases (Cluster AA1/AA4 in Table 5). Among these four genes, three of them are implicated in health disparities of African Americans. Specifically, gene *CRYAB* is related to dilated cardiomyopathy and myofibrillar myopathy. African Americans were found at higher risk for idiopathic dilated cardiomyopathy compared with Caucasian, and this could not be explained by income, education, alcohol use, smoking, or history of some other diseases (*Coughlin, Labenberg & Tefft, 1993*). Moreover, gene *CRYAA*, *CRYAB* and *CRYBB2* are all related to various types of cataract. It was reported that age-specific blindness prevalence was higher for African Americans compared with Caucasian, and cataract accounts for 36.8% of all blindness in African American, but for only 8.7% in Caucasian (*Congdon et al., 2004*).

**Table 5  Associated diseases of genes with enriched GO terms.**

| Cluster | Gene | Associated Disease |
|---------|------|--------------------|
| **AA1 and AA4** | *HSPB1* | Axonal Charcot-Marie-Tooth disease type 2F |
| | | Distal hereditary motor neuronopathy type 2B |
| | *CRYAA* | Multiple types of cataract 9 |
| | *CRYAB* | Multiple types of cataract 16 |
| | | Dilated cardiomyopathy-1II |
| | | Myofibrillar myopathy-2 |
| | | *CRYAB*-related fatal infantile hypertonic myofibrillar myopathy |
| | *CRYBB2* | Multiple types of Cataract 3 |
| **CA1** | *ATP2A1* | Brody myopathy |
| | *ATP2A2* | Acrokeratosis verruciformis |
| | | Darier disease |
| | *PLN* | Dilated cardiomyopathy-1P |
| | | Familial hypertrophic cardiomyopathy-18 |

**Notes.**

Only GO terms that contain CNV-genes are studied due to our focus on the role of CNV-genes in health disparity.

How could *HSPB1* duplication contribute to health disparities? Based on the direct interaction between *HSPB1* and *CRYAB* and the fact that both genes are expressed in Z-disc (Table 4), it is plausible that *HSPB1* may play an unknown role in cardiomyopathy. Alternatively, *HSPB1* might be involved in cataract, because *HSPB1*, *CRYAA* and *CRYAB* interact with each other and all can negatively regulate the apoptotic process (Table 4). Studies suggested that lens epithelial cell apoptosis may be a common cellular basis for initiation of non-congenital cataract formation (*Li et al., 1995*), and inhibition of epithelial cell apoptosis may be one possible mechanism that inhibits cataract development (*Nahomi et al., 2013*). Our results here argue for further experimental studies to test the possible role of *HSPB1* CNVs in cardiomyopathy or cataract/blindness in African Americans.

## Duplication of *ATP2A1* and cardiomyopathy

Gene *ATP2A1* in cluster CA1 is located in a genomic duplication region that occurs only in Caucasians (Table 3). We found that four genes in cluster CA1 are enriched with various GO terms that involve *ATP2A1* (Cluster CA1 in Table 4), and three of those four genes are related to diseases when they are mutated (Cluster CA1 in Table 5).

How would *ATP2A1* influence health disparities? Among the diseases related to the pathogenic genes in cluster CA1, idiopathic dilated cardiomyopathy occurs less often in Caucasians than in African Americans (*Coughlin, Labenberg & Tefft, 1993*). Based on the fact that *ATP2A1* interacts directly with *PLN*, and that they are both involved in the same biological processes and exist in the same cellular component (Table 4), it is plausible to suggest that duplication of *ATP2A1* may lead to the health disparity in idiopathic dilated cardiomyopathy. One possibility is that higher copies of *ATP2A1* may offer some benefits to Caucasians. Studies have shown that increased activity of sarco/endoplasmic reticulum $Ca^{2+}$-ATPase 1 (SERCA1), which is encoded by *ATP2A1*, can partially rescue the heart

from ·OH-induced injury (*Hiranandani, Bupha-Intr & Janssen, 2006*), and protect the heart from ischemia-reperfusion (I/R) injury (*Talukder et al., 2007*). Another possibility is that higher copies of *ATP2A1* only lead to moderate risk of cardiomyopathy in Caucasians, and this moderate effect is overshadowed by other genetics factors not covered by our CNV dataset.

## Remarks, limitations, and future directions

Although genetic factors play a crucial role in health disparities, only a few association studies have been reported in health disparities in common complex diseases, such as breast cancer (*Long et al., 2013*), prostate cancer (*Bensen et al., 2014*; *Bensen et al., 2013*; *Xu et al., 2011*), type 2 diabetes (*Ng et al., 2014*) and vascular diseases (*Wei et al., 2011*).

Our study here is closely related to network-based meta-analyses of GWAS results (*Atias, Istrail & Sharan, 2013*; *Leiserson et al., 2013*). One important aim of network-based meta-analysis of GWAS data is to distinguish the bona fide causal gene from other genes in the same haplotype block associated with the significant SNP. Likewise, our network approach aims to predict a potential causal gene from a population-specific CNV that can be associated with pathogenic genes.

Noticeably, our method does not require network permutations, whereas many existing methods of network/pathway based meta-analyses of GWAS data do. This difference is because we first partitioned the network into clusters and then performed association tests. In comparison, many network based GWAS meta-analysis methods use traversal distances to seed genes to evaluate candidate genes. This kind of traversal distance based method generally prohibits pre-partition of network into clusters and require network permutations for estimation of *p*-values. It can be seen that our cluster-based method naturally accommodates multiple candidate genes in the association analysis, whereas traversal distance in a network is by definition often limited to single candidate gene evaluation.

The clustering method of MCL that we chose has been consistently reported to work better than several other methods in detecting annotated protein complexes (*Pizzuti & Rombo, 2014*), is more tolerant to noises in the network datasets (*Vlasblom & Wodak, 2009*), and is argued to be the most reliable and robust method for network clustering analysis especially when interaction networks contain many noises and missing data (*Vlasblom & Wodak, 2009*; *Wang et al., 2010*). Nevertheless, this clustering procedure has introduced some limitations in our analysis.

The first major limitation is that the biological meanings of many clusters generated by MCL may be limited or ambiguous. For example, GO enrichment test suggested that cluster AA1 and AA4 are involved in visual perception and eye development, but GO term related to cardiomyopathy was not enriched. Moreover, gene–gene interactions (also known as edges) in our networks are unweighted. Since MCL basically partitions genes into strongly connected groups and separates these groups based on weak-flows (*Lin et al., 2007*), MCL essentially partitions networks only based on gene connection patterns (i.e., network topology) in this study. It is known that gene expression can be used to weight gene/protein interactions and thereby improve the biological relevance of

gene/protein networks (*Csermely et al., 2013*; *Liu & Chen, 2012*; *Liu et al., 2013*; *Qin & Yang, 2008*; *Wu, Zhu & Zhang, 2012*). In the present study, we were not able to use gene expression data sets relevant for health disparity-related diseases—a limitation that we hope to overcome in the future.

The second major limitation is the difficulty for parameter optimization due to uneven cluster sizes. Like other clustering methods, MCL yields clusters with uneven sizes, and makes it challenging for us to optimize the inflation parameter to find a level of sensitivity that can be acceptable in all cases. Consequently, significant associations were mostly detected when clusters sizes were moderate but not when cluster sizes were too large. We mitigated this problem to some extent by trying a range of values for the inflation parameter, but an optimal 'default' setting remains a challenge.

In future studies, we plan to address these limitations of the present study by integrating functional genomics data sets, such as gene expressions, into gene networks to generate weighted interactions, and by developing step-wise clustering methods.

## CONCLUSIONS

In this study, gene clusters were inferred from two human gene/protein networks, HPRDNet and MultiNet, by the MCL clustering algorithm with different parameters. Each cluster was ranked using the products of FDR values based on the right-tailed Fisher's exact tests for enrichment of pathogenic or CNV-genes. Five clusters were consistently found to be enriched with both pathogenic genes and genes located in African-American or Caucasian CNVs. In cluster AA1, AA2, AA3 and AA4, gene *HSPB1* is duplicated more frequently in African Americans. In clusters CA1, gene *ATP2A1* is duplicated only in Caucasians. All gene clusters are associated with certain diseases that occur more often in one population than in the other. Although we only studied population-preferred CNVs and did not consider the roles of other genetic factors, our computational studies have generated some interesting hypotheses for further experimental studies to understand health disparities in these diseases.

### List of Key Abbreviations

| | |
|---|---|
| **CNV** | Copy number variation |
| **SNP** | Single nucleotide polymorphism |
| **PPIN** | Protein–protein interaction network |
| **HPRD** | Human protein reference database |
| **PPI** | Protein–protein interaction |
| **AA** | African American |
| **MCL** | Markov Cluster Algorithm |
| **FDR** | False discovery rate |
| **GO** | Gene ontology |
| **OMIM** | Online Mendelian Inheritance in Man |
| **dbSNP** | Single Nucleotide Polymorphism Database |
| **SERCA1** | Sarco/endoplasmic reticulum $Ca^{2+}$-ATPase 1 |

## ACKNOWLEDGEMENTS

The authors thank three reviewers for constructive comments that have greatly improved the quality and presentation of this work.

### Funding

Yi Jiang and Li Yang were partially supported by the Tennessee Higher Education Commission's Center of Excellence in Applied Computational Science and Engineering. Hong Qin was partially supported by the Spelman Center for Health Disparities Research and Education (NIH 5P20MD000215-05) and the Spelman ASPIRE program (NSF award number 0714553). The funders had no role in study design, data collection and analysis, decision to publish, or preparation of the manuscript.

### Grant Disclosures

The following grant information was disclosed by the authors:
Tennessee Higher Education Commission's Center of Excellence in Applied Computational Science and Engineering.
Spelman Center for Health Disparities Research and Education: NIH 5P20MD000215-05.
Spelman ASPIRE program: NSF award #0714553.

### Competing Interests

The authors declare there are no competing interests.

### Author Contributions

- Yi Jiang performed the experiments, analyzed the data, wrote the paper, prepared figures and/or tables, reviewed drafts of the paper.
- Hong Qin conceived and designed the experiments, analyzed the data, wrote the paper, prepared figures and/or tables, reviewed drafts of the paper.
- Li Yang conceived and designed the experiments, wrote the paper, reviewed drafts of the paper.

### Supplemental Information

Supplemental information for this article can be found online at http://dx.doi.org/10.7717/peerj.677#supplemental-information.

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
