# Peer review of "Using network clustering to predict copy number variations associated with health disparities"

_PeerJ, doi:10.7717/peerj.677_

## Round 0.1 · original submission · Minor Revisions

Based on the reports on hand, I would like to suggest the minor revision. The author should revise the manuscript in the light of the reports. Particularly, the biological problem should be clarified, the clustering method should be tried in different settings and datasets. Furthermore, multiple testing should be considered in the calculation.

Reviewer 1 ·

Basic reporting

No comments

Experimental design

The question is not clearly defined. To my understanding, the authors integrated gene networks (HPRD and MultiNet), known pathogenic gene lists and differential CNVs between African Americans and Caucasians. But the question they want to answer is not clear. Generally, it is trivial to obtained clusters enriched with either pathogenic genes or CNVs. Although Fisher's exact test was applied to evaluate the statistical significance of the enrichment, I do not follow the validity of this hypothesis testing method. In my opinion, random networks constructed based on the original gene networks should be better random controls. Even statistically significant clusters enriched with pathogenic genes or CNVs were found, the biomedical meanings of these clusters are still unclear. The authors applied GO enrichment analysis to highlight the biological meanings of the clusters. However, GO enrichment analysis is independent to the previous enrichment analyses. The relationship of GO enrichment analysis and pathogenic gene enrichment should be clarified clearly.

Validity of the findings

Because the clusters the author identified are heavily dependent on the clustering algorithms and parameters, the findings are vulnerable. More clustering algorithms and parameters should be tested and the criteria to select clustering schemes should be defended.

Reviewer 2 ·

Basic reporting

In this work, the authors performed a systematic investigation of CNVs in the network space. They inferred network clusters that are population specific, suggesting CNVs’ potential roles in health disparities. This work is quite novel and interesting. The results are robust. The manuscript is written well.

Experimental design

The scientific question is clear. Population specific CNVs are used in network clustering analysis. Two networks are used. The design is robust.

Validity of the findings

Results are valid through different networks and multiple datasets. The statistical tests are appropriate.

Additional comments

The work is high quality. I only have a few minor comments that I believe the authors can take care in finalizing the manuscript.

Table 2, gene symbols should be italicized. The coordinates better to have corresponding reference genome assembly number. And it is better not to have so many decimals in frequency values.

Table 3: GO Type -> GO Domain and should have full name (e.g. Molecular Function) or abbreviation (e.g. MF); Go term: it is better to provide GO ID since may GO term names look similar.

Table 3: Gene symbols should be italicized.

Table 4: not sure those numbers after disease name. What are those 2F, IIB, 2, 7, 27, 11, 1P, 18?

The authors may check the data and cite this CNV resource: PubMed ID: 24244640; http://www.ncbi.nlm.nih.gov/pubmed/24244640

Reviewer 3 ·

Basic reporting

No Comments

Experimental design

No Comments

Validity of the findings

No Comments

Additional comments

In this paper, Jiang and colleagues proposed a network-based method to identify the associations between copy number variations and health disparties. In a word, I support its publication if the authors make some revisions. The following lists my major comments.

1. The network clustering. The authors implemented a Markov clustering method to group the documented protein-protein interaction into small protein sets. Some biases might exist in this process. The PPI network has no specific information for the conditions or phenotypes, which just collected the binary links between proteins, i.e., the interaction does not contain a metric to measure their associations with the conditions such as health disparties. The MCL methods can also be used to group weighted graphs. The gene network should be conditioned before the prediction. The authors should clarify this issue. Moreover, the interactions in the two databases contain many noise, the network should be justified firstly.

2. The comparison study with GWAS study. The paper proposed a network-based method for identifying these pathogenic genes. Comparison will be interesting to the readers for the difference between the traditional GWAS identification and the proposed method. The P-values listed in Table 3 should be corrected for multiple testing and the rank tests should also consider their FDR.

3. The following important references in this direction should be cited (PMID:22729399 for PPI; PMID:24067414 for gene network, PMID:22360268 for network-based disease analysis; and PMID:14735121 for network biology)

---

## Round 0.2 · Minor Revisions

I suggest you add some discussion about the limitations mentioned by Reviewer 1 in any further revision.

---

## Round 0.3 · accepted · Accept

The manuscript has been improved after revision. I suggest its acceptance.